# Effect of Naphthalene-Based Superplasticizer and Polycarboxylic Acid Superplasticizer on the Properties of Sulfoaluminate Cement

**DOI:** 10.3390/ma14030662

**Published:** 2021-01-31

**Authors:** Yonghua Wu, Qiqi Li, Guoxin Li, Shiying Tang, Mengdie Niu, Yangfan Wu

**Affiliations:** 1College of Materials Science and Engineering, Xi’an University of Architecture and Technology, Xi’an 710055, China; wuyonghua@xauat.edu.cn (Y.W.); liqiqijobmail@163.com (Q.L.); niumengdie@xauat.edu.cn (M.N.); 18829502569@163.com (Y.W.); 2Northwest Institute of Nuclear Technology, Xi’an 710024, China; tangshiyingmail@126.com

**Keywords:** superplasticizer, sulphoaluminate cement, compressive strength, ettringite

## Abstract

In order to study what the effect of superplasticizers on the setting time, fluidity and compressive strength of calcium sulfoaluminate cement (CSA) a naphthalene-based superplasticizer (BNS) and a polycarboxylic acid superplasticizer (PC) were selected to interact with CSA pastes and ye’elimite, respectively. X-ray diffraction (XRD), thermogravimetric (TG) analysis and scanning electron microscopy (SEM) analytical methods were used to investigate the class, amount and microstructure of the CSA pastes and ye’elimite pastes hydration products under the effect of the superplasticizers. The results showed that the addition of BNS can promote ettringite generation and thus improve the early compressive strength. As the addition of BNS increased from 0.8 wt% to 2.0 wt%, the initial setting time was prolonged 10 min, the final setting time was prolonged 7 min, the 5 min fluidity was improved from no fluidity to 220 mm. However, as the addition of PC increased from 0.08 wt% to 0.20 wt%, the setting time of the PC just changed within 3 min; the 5 min fluidity increased from 110 mm to 195 mm and no 15 min fluidity at all was observed. AS seen by SEM, it can be stated that generated ettringite under the addition of PC was layered and lacking bonding, and its morphology changed from rod-like to flake-like, leading to a decrease in early compressive strength.

## 1. Introduction

Calcium sulfoaluminate cement (CSA), due to its properties of high early strength, short setting and hardening time, impermeability, lower CO_2_ emissions and less energy required for its production, is widely used as a repair material at present [1,2]. The main hydraulic phase of CSA is ye’elimite (4CaO·3Al_2_O_3_·SO_3_, C_4_A_3_S), with lower amounts of belite (C_2_S), anhydrite (CS), and gehlenite (C_2_AS) also being present [3,4]. The main hydration reactions of CSA are as follows in Equations (1) and (2).

(1)$$C_{4}A_{3}\bar{S}\;+2C\bar{S}\;\cdot {2H+34H=C}_{3}A\cdot 3C\bar{S}\;\cdot {32H+2\mathrm{AH}}_{3}$$

(2)$$C_{4}A_{3}{\bar{S}\;+18H=C}_{3}A\cdot C\bar{S}\;\cdot {12H+2\mathrm{AH}}_{3}$$

Generally, ye’elimite involves anhydrite and then generates ettringite (C_3_A·3CS·32H, AFt) and aluminum hydroxide (AH_3_). Certainly, hydration of ye’elimite gives rise to monosulfoaluminate without anhydrite [5,6,7,8,9]. Ettringite, the main hydration product of CSA, which is often formed rapidly and then the setting time shortened and the fluidity decreased to provide support for the early strength within a few hours. Generally, the ettringite morphology involves fine needles, short columns, dendrites and round rods, etc. [10,11]. However, on account of the uneven adsorption or bonding on the surface of ettringite, ions in the chemical environment may preferentially adhere to certain faces of the ettringite crystals and change their nucleation and growth [12,13,14]. Whether the change of ettringite formation environment has a direct effect on the macroscopic properties of cement-based systems with ettringite as the main product is not yet known. Previous studies showed that the characteristics of ettringite indeed are related to the chemical environment [15,16,17]. The morphology of ettringite in the presence of over 300 chemicals and admixtures was very different. Moreover, most of them are present in the concrete environment [18]. Superplasticizers are the most widely used surfactants, containing several functional groups in the cement hydration system, which is an essential component that can change the chemical environment [19].

The formation of ettringite is seriously affected by the chemical environment. In the solution chemical environment added naphthalene-based superplasticizer delayed the formation of ettringite. With the increase of the content of naphthalene-based superplasticizer, not only did the size of ettringite decrease, but also the micromorphology of ettringite changed from the original thick rod shape to a fibrous shape [20]. Prince et al. found that mixtures in which no superplasticizer was added contain very thin needles of ettringite, which connect the initial granular particles. The paste incorporating naphthalene-based superplasticizer showed an amorphous background with some massive clusters of crystallite [21]. Similarly, when sodium benzenesulfonate, sodium dodecylbenzenesulfonate and melamine-based superplasticizer were added in different forms to form ettringite, these additives all delayed the hydration of the hydration phase, inhibited the growth of ettringite crystals and led to a decrease in the amount of ettringite produced in the early stages [22,23,24].

However, there are opposite conclusions. For example, in the environment of addition of melamine-based superplasticizer, naphthalene-based superplasticizer and lignin-based superplasticizer, all three superplasticizers can accelerate the crystallization of ettringite in early strength cement [25]. Ramachandran et al. showed that melamine-based superplasticizer or naphthalene-based superplasticizer can delay the dissolution of the hydration phase, but can accelerate the reaction to form ettringite [26]. X-ray diffraction (XRD) was used to study the effect of melamine-based superplasticizer on the hydration of ettringite at different dosages. The agent accelerated the hydration, leading to the acceleration of ettringite crystallization [27]. Colombo et al. studied the effect of calcium lignosulfonate addition on the types and morphologies of cement hydration products, ettringite presented a compact cuboid shape instead of needle-like shape, and the morphology of ettringite crystals has not changed with the addition of superplasticizer [28]. These results are not completely consistent with the findings of other studies in the prior literature.

At present superplasticizer is the most common admixture used in modern cement and concrete construction. Likewise, it is widely used in CSA. To sum up above, the effect of superplasticizers on CSA is different distinctly. Hence, the significance of research is figuring out what influences CSA under the action of superplasticizers and how to apply CSA better in real projects.

Herein, a naphthalene-based superplasticizer and a polycarboxylic acid superplasticizer commonly used in daily engineering construction were selected and the effects of the superplasticizers on the setting time, fluidity, and compressive strength of CSA were studied. We focus on the process of hydration of C_4_A_3_S, the main mineral of CSA, to ettringite, and the effect of superplasticizers on the amount and morphology of ettringite to verify the relationship between the macro-properties and micro-products. Moreover, the class, amount, and microstructure of hydration products, especially for ettringite, were researched by XRD, thermogravimetric (TG), and scanning electron microscopy (SEM), respectively.

## 2. Materials and Methods

### 2.1. Materials

Rapid hardening sulphoaluminate cement with strength grade of 42.5 and standard sand were used in the present study. The two types of common superplasticizer used were a powdered naphthalene-based superplasticizer (BNS) (provided by Shaanxi Rising Building Material Technology Co., Ltd., Xi’an, China) and a powdered polycarboxylic acid superplasticizer (PC) (provided by Shaanxi Kezhijie New Material Co., Ltd., Xi’an, China). Especially, during the sample preparation, the powery superplasticizer should be mixed with cement first before adding water. The dosage of superplasticizer was determined as the weight percentage of superplasticizer in the binders. Calcium carbonate gypsum (CaSO_4_·2H_2_O) and aluminum hydroxide (Al(OH)_3_) all provided by Sinopharm Chemical Reagent Co., Ltd. (Xi’an, China) with purity of 99.9% were used. Absolute ethyl alcohol with purity of 99.7% was provided by Xi’an Chemical Co., Ltd. (Xi’an, China).

### 2.2. Synthesis and Characterisation of C_4_A_3_S

According to the molecular formula of C_4_A_3_S (4CaO·3Al_2_O_3_·SO_3_), calcium carbonate (CaCO_3_), gypsum (CaSO_4_·2H_2_O) and aluminum hydroxide (Al(OH)_3_) were mixed by a high speed mixer (Model-U400/80-220, Shanghai Weite Motor Co., Ltd., Shanghai, China) in the proportions listed in Table 1 to form the raw material, and mixed with a certain amount of deionized water under stirring for 2 h and then dried in an oven (Model-101-2AB, Tianjin Test Instrument Co., Ltd., Tianjin, China) at 105 °C for use. The raw material was pressed in alumina crucibles and placed in a high-temperature furnace (Model-CSL-16-12Y, Sinosteel Luoyang Refractory Research Institute Co., Ltd., Henan, China) to be fired as heating system shown in Table 2. After the firing, it was taken out at 800 °C and quenched to room temperature. The purity of the clinker was checked by the XRD test method and the result is shown in Figure 1. According to the standard diffraction pattern, it can be seen the interplanar spacing d value in Figure 1 is consistent with target substance ye’elimite, which indicates that ye’elimite is successfully synthesized.

### 2.3. Sample Preparation

To ensure CSA pastes has good workability and no serious bleeding during testing many times the state of CSA pastes mixed with different dosage of superplasticizers was examined and then the suitable dosage range of BNS was determined to increase from 0.8 wt% by 0.3 wt% to 2.0 wt%; the suitable dosage range of PC was determined to increase from 0.08 wt% by 0.03 wt% to 0.20 wt%. CSA mortar samples of 40 mm × 40 mm × 160 mm, with the same water to cement weight ratio 0.35 as CSA pastes and cement to sand weight ratio 1, were made of calcium sulfoaluminate cement, sand, water and superplasticizers. The specific processing steps were to mix all components stirring at low speed for 30 s in a cement mortar mixer (Wuxi Xiding Construction Engineering Instrument Factory, Wuxi, China), adding water at 10 s during the first 30 s, and then putting standard sand into the mixer at the second 30 s mark under low speed, and then turning to high speed stirring for 30 s again and stopping 90 s, and mixing everything for 60 s at high speed again. When the mixing program was completed, CSA mortar samples were cast into the mold twice and vibrated on a vibrating table (Wuxi Xiding Construction Engineering Instrument Factory) for 30 s.

In order to obtain preferable workability and satisfy more water demand, the water to binder weight ratio of C_4_A_3_S was 0.38. C_4_A_3_S mortar samples were cast into 20 mm × 20 mm × 20 mm shapes with a binders to sand weight ratio of 1. The binders consisted of C_4_A_3_S, gypsum, calcium carbonate and superplasticizers. The ratio of gypsum to C_4_A_3_S was 0.56. Calcium carbonate was used for maintaining the stability of ettringite with calcium carbonate to C_4_A_3_S was 0.39. In particular, the hydration speed of C_4_A_3_S was faster than that of the cement, hence, the mixing time should be adjusted to meet the practical situation under the referrence of CSA mortar samples processing steps.

All samples were cast at 20 ± 2 °C, and then cured in a standard curing chamber at the temperature of 20 ± 2 °C and the relative humidity of 95% HR until 2 h, 1 d, 7 d, and 28 d to test their compressive strength with universal testing machine (Wuxi Xiyi Building Material Instrument Factory).

### 2.4. Test Methods

#### 2.4.1. Setting Time

The setting time of CSA pastes with different dosages superplasticizers were measured by a Vicat apparatus (Wuxi Building Material Instrument Machinery Factory) in accordance with the ISO 9597: 2008. The setting situation of specimens should be tested every 30 s after the addition of water to the mixes began. Every sample should be tested three times and the results averaged.

#### 2.4.2. Compressive Strength

The compressive strength of CSA mortars were measured by a universal testing machine (Wuxi Xiyi Building Material Instrument Factory) following ISO 679: 2009 ‘‘Cement Test Methods Determination of Strength”. The specimens were tested at the age of 2 h, 1 d, 7 d, and 28 d after mixing with water. Every sample should be tested six times and the results then averaged.

#### 2.4.3. Fluidity

The fluidity of CSA paste was tested in accordance with the Chinese Standard GB/T 8077-2012 ‘‘Methods for testing uniformity of concrete admixture”. The specimens were casted into a conical mold (with a height of 60 mm, top diameter of 36 mm, and bottom diameter of 60 mm). Then the conical mold was lifted up to let the paste flow freely for 30 s onto a glass plate [29]. The fluidity was determined by the average of diameters at two vertical directions. The fluidity of the cement pastes was measured repeatedly at 5 min and 15 min after mixing to record the fluidity loss.

#### 2.4.4. XRD Analysis

CSA pastes and C_4_A_3_S pastes were prepared for X-ray diffraction (XRD) analysis. At 10 min, 2 h, 1 d, and 7 d after mixing with water, specimens were ground into powder sieved to 75 μm and their hydration were blocked by absolute ethyl alcohol, followed by being dried in a vacuum dryer (Model-DZF, Beijing Kewei Yongxing Instrument Co., Ltd., Beijing, China) at 40 °C for 48 h, finally sealed there till XRD analysis. XRD patterns of pastes at the given time were observed in an X-ray diffractometer (Model-D/MAX-3C, Rikagu, Tokyo, Japan) with Cu Kα radiation.

#### 2.4.5. TG Analysis

CSA pastes and C_4_A_3_S pastes were prepared for thermogravimetric (TG) analysis. The treatment process for specimens was same as the XRD analysis. Approximately 10 mg of the specimens were placed in open Al_2_O_3_ pans. TG analysis was performed from 25 °C to 900 °C with a heating rate of 10 °C/min under N_2_, using thermogravimetric analyzer (Model-SDTQ 600, Netzsch, Bavaria, Germany).

#### 2.4.6. SEM Analysis

The microstructure of CSA pastes and C_4_A_3_S pastes were observed by scanning electron microscopy (SEM, Model S-4800, Hitachi, Tokyo, Japan). The treatment process for specimens was same as the XRD analysis except for powder replaced by particles about 5 mm. In order to observe clearly, samples were sputtered by ion sputtering (Model-E-1045, Hitachi) with gold for 45 s.

## 3. Results and Discussion

### 3.1. Setting Time

The setting time of BNS and PC superplasticizers with different dosages are displayed in Table 3. Obviously, with the increase of the amount of BNS superplasticizer, the setting time as a whole was prolonged, and the most obvious effect is when the amount was 1.7%, initial setting reaching 32 min and 39 min for final setting. This could be explained by the fact that more superplasticizer is adsorbed on the surface of cement particles as the amount of naphthalene-based superplasticizer increases, then more free water is released so the setting time is prolonged under the effect of electrostatic repulsion.

Different from the BNS superplasticizer, the setting time did not increase significantly when the PC superplasticizer was added. In the dosage ranges from 0.08% to 0.20%, the initial and final setting time varied within 3 min.

### 3.2. Fluidity

The 5 min and 15 min fluidity of BNS and PC superplasticizers with different dosages are shown in Table 4. The pastes had no 5 min and 15 min fluidity without superplasticizer. When added with 0.8 wt% BNS, the pastes had no fluidity at 5 min and 15 min. As the amount of BNS increased from 1.1 wt% to 2.0 wt%, the 5 min fluidity improved from only 90 mm to 220 mm. However, the CSA pastes had no 15 min fluidity until the BNS dosage increased to 1.7 wt%. The fluidity at 15 min increased from 100 mm to 135 mm, and the 5 min fluidity increased from 205 mm to 220 mm when the BNS dosage grew from 1.7 wt% to 2.0 wt%. The fluidity loss between 5 min and 15 min decreased. It is also can be seen in the Table 4, the 5 min fluidity of PC superplasticizer increased from 110 mm to 195 mm when the added dosage was increased from 0.08 wt% to 0.20 wt%. There was no fluidity of 15 min for all dosages. Therefore, the fluidity loss between 5 min and 15 min was keeping increasing at PC dosage ranges.

Current studies generally show that this loss of fluidity over time is mainly caused by cement hydration [1,2,5]. On the one hand, hydration products are produced in large amounts in the liquid phase on the surface of the particles and as the network structure of the coagulation and crystallization hydration products grows, thus causing the loss of fluidity of the cement slurry over time. On the other hand, the formed hydration products covering on the adsorption layer would also greatly reduce the effect of electrostatic repulsion [30], resulting in a rapid reduction of the zeta potential of the system. Macroscopically, the fluidity of freshly mixed cement slurry is lost over time too fast. In this study the loss of fluidity in addition of PC superplasticizer may be caused by ettringite produced rapidly resulting from special molecular structure of PC superplasticizer and its effect of steric hindrance.

### 3.3. Compressive Strength

CSA mortars compressive strength involved with superplasticizers of BNS and PC are displayed in Figure 2. As shown in Figure 2a, with the increase of the amount of BNS superplasticizer, the compressive strength of CSA mortar showed an increasing trend in all ages. Focusing on the early strength, it can be found that when the BNS dosage was 1.4%, the 2 h strength increase reached a maximum of 17%. The compressive strength of CSA mortar mixed with PC superplasticizer was more complicated. As shown in Figure 2b, according to the strength curve of 2 h, the PC content ranges from 0.08% to 0.17%, and the compressive strength of CSA mortar increased slightly. The compressive strength of CSA mortar mixed with PC superplasticizer at all ages has one thing in common. When the PC content reached 0.20%, the compressive strength decreased significantly and it was much lower than the minimum compressive strength of 0.08%, especially at 2h the compressive strength decreased by 36%.

In order to study further the effect between dosages and types of superplasticizers and CSA, C_4_A_3_S mixed with minimum and maximum dosage of BNS and PC were selected to be researched. And the results of C_4_A_3_S mortars compressive strength with and without superplasticizers are shown in Table 5. Obviously, compared to C_4_A_3_S mortars compressive strength without superplasticizers, the compressive strength of C_4_A_3_S at 1d and 7d both increased whether the BNS superplasticizer was 0.8% or 2.0%. In contrast, the C_4_A_3_S mortars compressive strength of the PC superplasticizer decreased at maximum content of 0.20%.

The system of hydration of CSA and C_4_A_3_S mortars all would be more homogeneous because more superplasticizer is adsorbed on the surface of cement particles. Then the dispersion of BNS superplasticizer improved and the system generated more ettringite as TG results shown in Table 6 and Table 7. The system with more ettringite showed higher compressive strength. In addition, the research showed that ettringite precipitation predominantly occurs via the production of more individual surface-bound needles rather than the existing needles increasing in thickness and length [31]. Hence, the precipitation and growth method of ettringite also need to be considered to study properties changes of hydration system. The compressive strength decreased badly when the dosage of PC superplasticizer increased from 0.08% to 0.20%. Therefore, this could be explained by that hydration products generated so fast that ettringite only layered stacking simply and lacked bonding. And the amount of ettringite in CSA pastes with PC superplasticizer at dosage of 0.20% was really more than 0.08% as shown in Table 6, which confirmed the assumption above.

### 3.4. XRD Analysis

To understand the superplasticizers’ effect on the hydration of cement in early times, XRD patterns of the hydration products of CSA at 2 h were investigated. The results are as shown in Figure 3, where the products hydrated at 2 h with BNS superplasticizer were ettringite, ye’elimite, anhydride, and calcium silicate, among which the peaks of ettringite were the densest. With the increase of BNS superplasticizer, the anhydrite decreased in the peak value indicating the degree of CSA hydration intensifies and the reaction became more complete. Similarly, the addition of PC superplasticizer did not produce new hydration products. According to Figure 3b, at 2 h the hydration products still consist of ettringite, ye’elimite, anhydride and calcium silicate, and the peak of ettringite increased as the dosage was upped.

Moreover, C_4_A_3_S hydrated with BNS and PC superplasticizers were tested by XRD at 10 min, 2 h, 1 d, and 7 d after mixing, the results are displayed in Figure 4. It can be seen from it that at the age of 10 min, no matter what the dosages of BNS and PC were, gypsum, C_4_A_3_S, calcite did not give good hydration. The peak of ettringite appeared from 2 h, and the peak of ettringite increased with age prolonged. Above all things, there were no other new crystalline phases except ettringite.

The results indicate that no new hydration phases produced with the further addition of superplasticizers. The characteristics of ettringite is the reason that caused differences in the hydration system of CSA with BNS and PC superplasticizer. This is discussed in the next sections.

### 3.5. TG Analysis

In order to know the content of given hydration products in specimens. The TG and differential thermogravimetric (DTG) curves of CSA pastes and C_4_A_3_S with different dosage of BNS and PC are shown in Figure 5 and Figure 6, respectively. The DTG curves show the existence of endothermic peaks at 50–150 °C, which are related to the dehydration of lattice water of ettringite. The ettringite content was calculated between 50 °C and 15 °C from the TG data using Equation (3) [32]:(3)$$\mathrm{AFt}\;(\%)=\frac{M_{L}}{0.35}$$

M_L_: Loss of weight of samples at 50–150 °C.

The weight loss curve of the superplasticizer and the CSA pastes was plotted by thermal analysis and results are shown in Figure 5. The ettringite content is shown in Table 6. When BNS superplasticizer was added to the CSA pastes with dosages of 0.8% and 2.0%, a higher amount of ettringite was formed after 2 h with a dosage of 2.0%, which was consistent with the ettringite amount of C_4_A_3_S hydrated with BNS superplasticizer in Table 7. More ettringite really provided higher compressive strength, which can be confirmed from Figure 2 and Table 5. Specifically, the more addition of BNS superplasticizer was adsorbed on C_4_A_3_S and the effect of dispersion make the reaction more complete and homogeneous and then promoted more ettringite amount to obtain higher compressive strength.

Also according to Table 6, the amount of ettringite stayed almost the same at 7 d and 28 d even if the dosage of PC superplasticizer was increased. Only at the age of 2 h there was 28% increment of ettringite when the dosage was increased from 0.08% to 0.20%. Nevertheless, the compressive strength of CSA pastes with PC superplasticizer at 2 h decreased by 36%. Similarly, the weight loss curve of C_4_A_3_S with and without superplasticizer samples is shown in Figure 6 and the ettringite amount calculated from the ettringite water loss range is shown in Table 7. The addition of PC superplasticizer with maximum dosage 0.2% at 2h early times producted more ettringite than the minimum dosage of 0.08%. However, as can be seen from Table 5 the 1 d and 7 d compressive strength decreased 25.9% and 22.5%, respectively, when the PC superplasticizer dosage increased from 0.08% to 0.20%. Ettringite content increased but strength decreased, which could be explained by the previous assumption that ettringite was generated very fast but underwent layered stacking. A lack of bonding is the reason why the compressive strength decreased.

### 3.6. SEM Analysis

The morphology and size of early hydrates at 2 h in presence of superplasticizer was investigated with SEM. CSA pastes and C_4_A_3_S pastes containing maximum and minimum dosages of BNS and PC superplasticizer, respectively, were studied. The hydrates of C_4_A_3_S formed in these samples were compared to those formed in reference samples without superplasticizers.

SEM images of CSA pastes with dosage of 0.8% BNS superplasticizers for 2 h is as shown in Figure 7a, flake-like and rod-like ettringite covering many gels forming a compact crystal structure. When the dosage of BNS increased from 0.8% to the maximum dosage of 2.0%, it can be seen from Figure 7b that a large number of flake-like and rod-like ettringite particles were more densely packed, which is also the reason for the increased compressive strength of the system. As shown in Figure 7c, when the PC dosage was minimum 0.08%, a large number of needle-like rod-shaped ettringites were alternately formed, and a large number of non-growing crystal particles were attached. When the amount was increased to 0.2%, rod-shaped ettringite was still visible in Figure 7d, and the system was denser, but the compressive strength reduced compared to the minimum dosage 0.08%.

Hence, in order to further know whether the addition of superplasticizer caused changes in hydrates morphology or not, Ettringite hydrated by C_4_A_3_S with maximum and minimum dosage of BNS and PC superplasticizer at 2 h were studied by SEM and the results are shown in Figure 8.

The results shown in Figure 8a–c were consistent with the previous description. It can be seen from them that when the 2 h BNS dosage was increased from 0.8% to 2.0%, in addition to the observation of unhydrated particles, rod-shaped ettringite was formed. A large number of crystal particles were attached to the surface, and the form of ettringite was better with an increase of BNS, which indicates that BNS with the dosage of 2.0% made the particles more dispersed and hydrated faster and produced more ettringite as identified in the TG analysis results. The addition of PC superplasticizer changed the morphology of ettringite, which is verified in Figure 8d,e. A large amount of flake-like ettringite is layered in Figure 8e. Firstly, in the previous XRD analysis of Figure 4c,d, it was found that the addition of PC did not produce any new crystalline material except ettringite. Secondly, in the TG analysis, it was found that the amount of ettringite was not reduced by the dosage of PC. According to Yan’s SEM study of ettringite [33], different crystal morphology of ettringite had different properties Therefore, the decrease in strength is attributed to the change in the shape of ettringite. The formation and growth of ettringite has been changed from rod-like to flake-like under the addition of PC. Ettringite with stacking morphology led to decreases in compressive strength.

## 4. Conclusions

(1)As the addition of BNS increases from 0.8 wt% to 2.0 wt%, the initial setting time is prolonged 10 min and the final setting time is prolonged 7 min. However, as the addition of PC increaes from 0.08 wt% to 0.20 wt%, the setting time of the PC just changed within 3 min.(2)The 5 min fluidity improved, from no fluidity to 220 mm as the addition of BNS increased from 0.8 wt% to 2.0 wt%. With the addition of PC increasing from 0.08 wt% to 0.20 wt%, the 5 min fluidity increased from 110 mm to 195 mm and no 15 min fluidity at all.(3)The addition of BNS can promote ettringite generation and then improved the early compressive strength. Moreover, the 2 h amount of ettringite in CSA pastes increased 5 wt% when the PC dosage was increased from 0.08 wt% to 0.20 wt%, which caused decrease in early compressive strength instead.(4)The properties of sulfoaluminate cement were affected by the changes in morphology of ettringite, which results from the addition of BNS and PC. The morphology of ettringite observed by SEM changed from rod-like to flake-like under the addition of PC. The generated flake-like ettringite stacked lacking bonding, causing a decrease in compressive strength.

## Figures and Tables

**Figure 1 materials-14-00662-f001:**
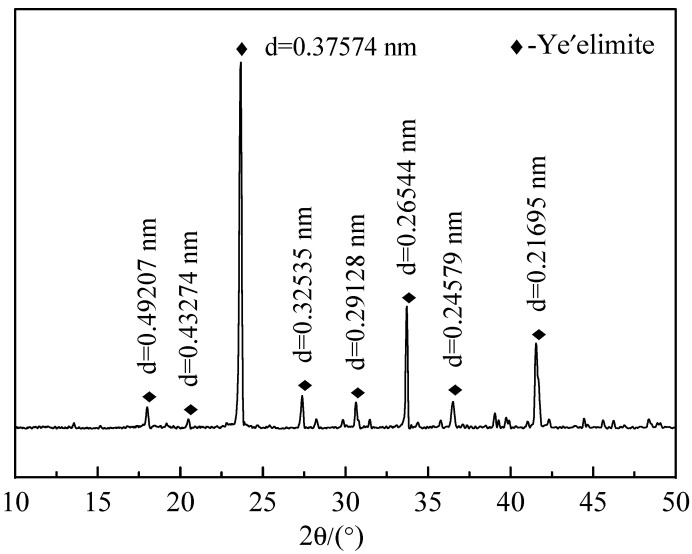
The XRD pattern of synthesized ye’elimite.

**Figure 2 materials-14-00662-f002:**
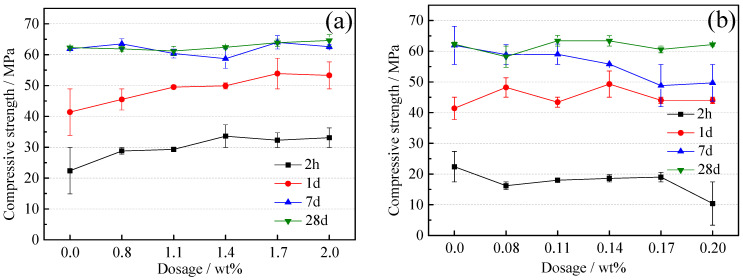
CSA mortars compressive strength with superplasticizers of (**a**) BNS, (**b**) PC.

**Figure 3 materials-14-00662-f003:**
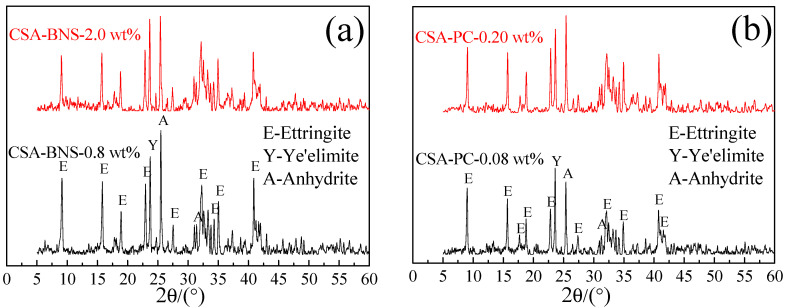
XRD patterns of CSA pastes at 2 h with maximum and minimum dosage superplasticizers of (**a**) BNS, (**b**) PC.

**Figure 4 materials-14-00662-f004:**
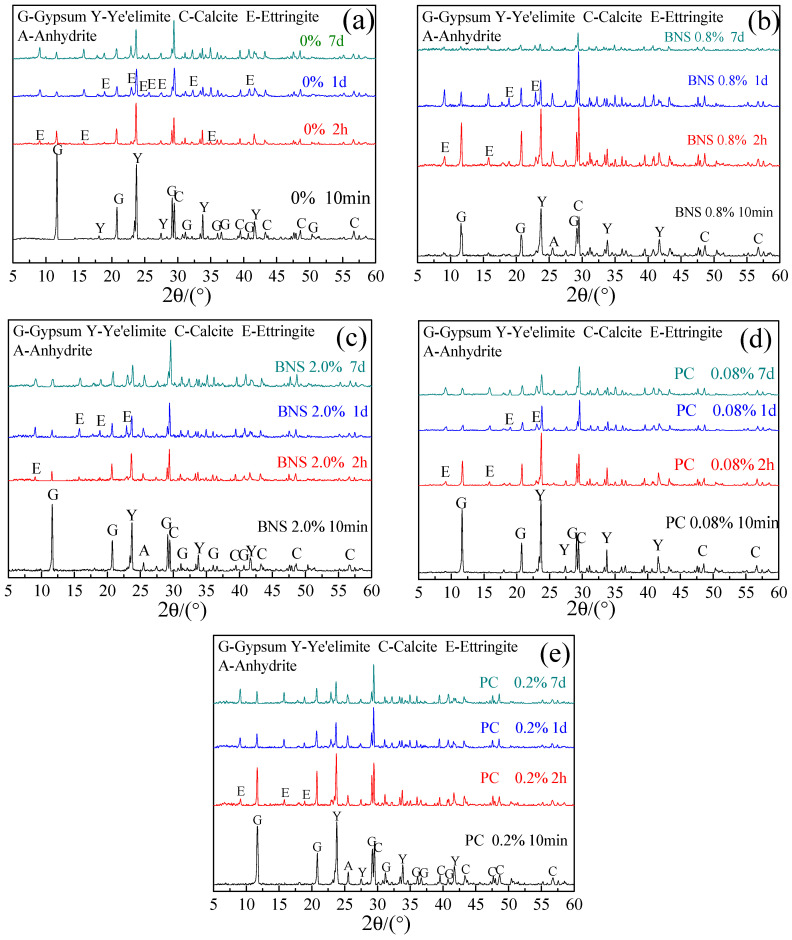
C_4_A_3_S pastes XRD patterns of (**a**) without superplasticizers, (**b**) BNS-0.8%, (**c**) BNS-2.0%, (**d**) PC-0.08%, (**e**) PC-0.20% with 10 min, 2 h, 1 d, 7 d.

**Figure 5 materials-14-00662-f005:**
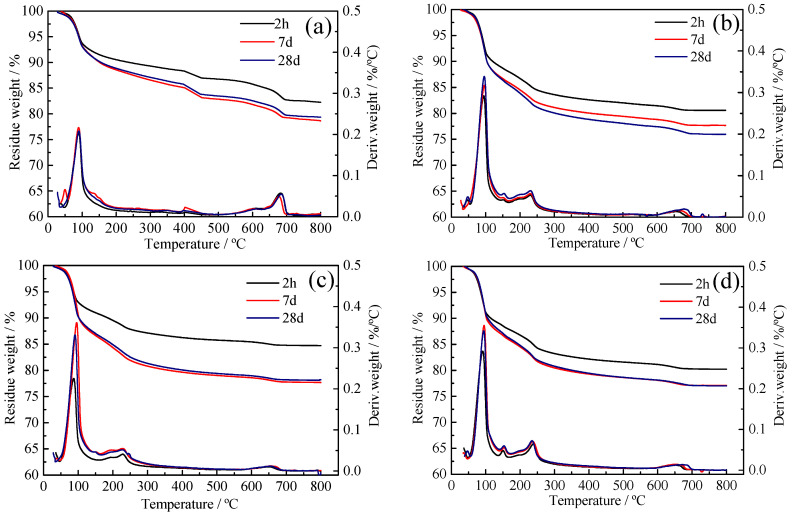
TG curves of CSA pastes with superplasticizers for 2 h, 7 d, 28 d (**a**) BNS-0.8%, (**b**) BNS-2.0%, (**c**) PC-0.08%, (**d**) PC-0.20%.

**Figure 6 materials-14-00662-f006:**
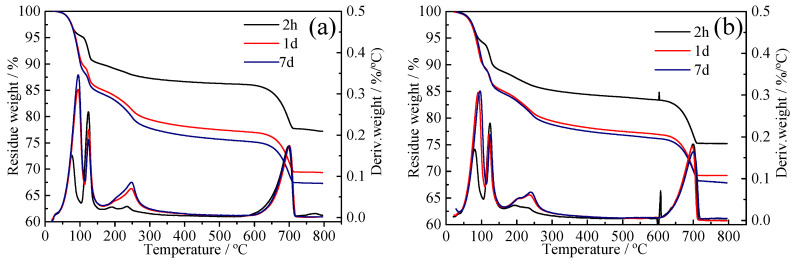
TG curves of C_4_A_3_S pastes without superplasticizers and without superplasticizers for 2 h, 1 d, 7 d (**a**) without superplasticizers, (**b**) BNS-0.8%, (**c**) BNS-2.0%, (**d**) PC-0.08%, (**e**) PC-0.20%.

**Figure 7 materials-14-00662-f007:**
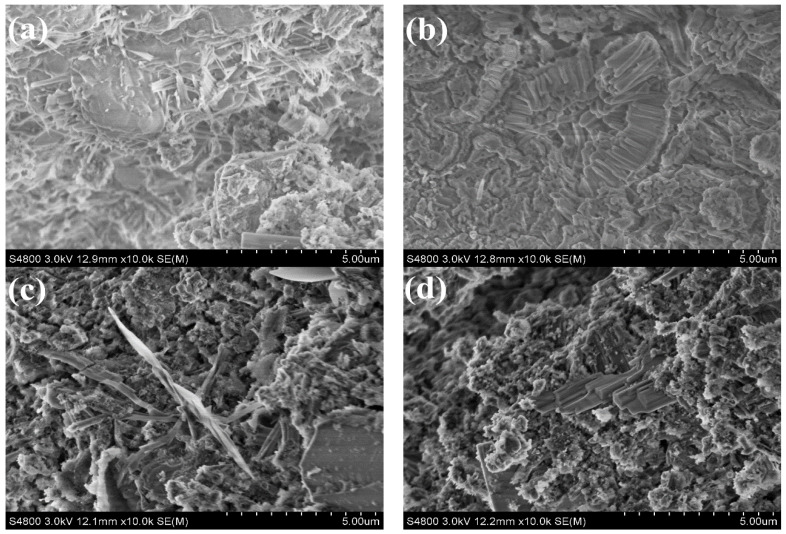
SEM images of CSA pastes with superplasticizers for 2 h (**a**) BNS-0.8%, (**b**) BNS-2.0%, (**c**) PC-0.08%, (**d**) PC-0.20%.

**Figure 8 materials-14-00662-f008:**
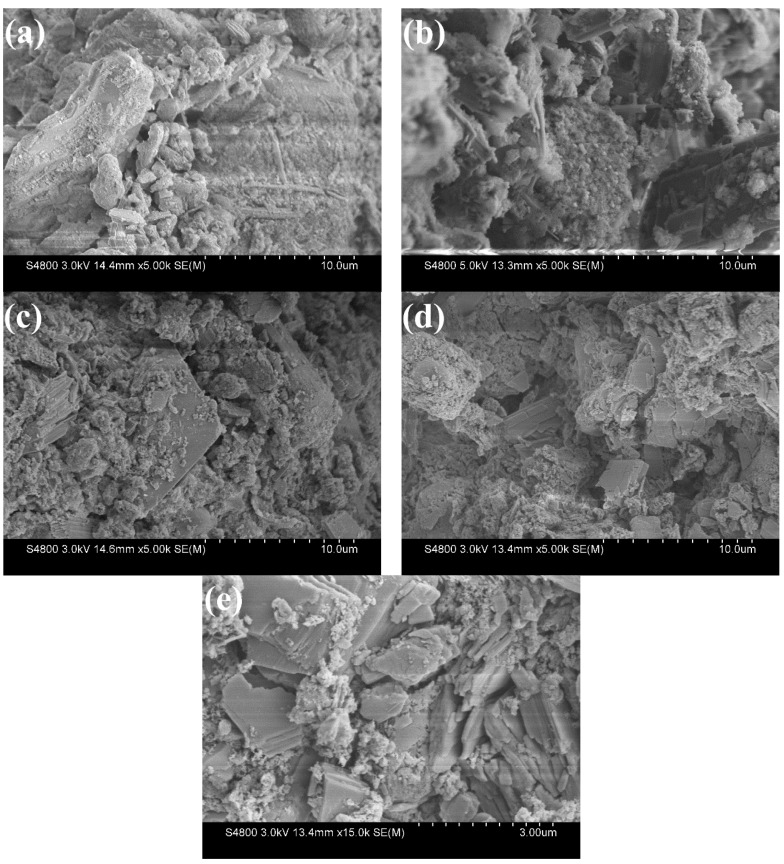
SEM images of C_4_A_3_S pastes without superplasticizers and with superplasticizers for 2 h (**a**) without superplasticizers, (**b**) BNS-0.8%, (**c**) BNS-2.0%, (**d**) PC-0.08%, (**e**) PC-0.20%.

**Table 1 materials-14-00662-t001:** Mix proportion of C_4_A_3_S raw material.

**Components**	CaSO_4_·2H_2_O	CaCO_3_	Al(OH)_3_
**Mix Proportion**	1.0	1.2	2.1

**Table 2 materials-14-00662-t002:** Heating system for synthesizing C_4_A_3_S.

Temperature/°C	Time/min
300	30
900	120
1300	150
1300	120

**Table 3 materials-14-00662-t003:** Setting time of BNS and PC superplasticizers with different dosages.

Superplasticizer/wt%	Setting Time		Superplasticizer/wt%	Setting Time	
	Initial/min	Final/min		Initial/min	Final/min
0	15 ± 1.53	21 ± 0.58	PC-0.08	22 ± 0.58	27 ± 1.00
BNS-0.8	20 ± 0.58	28 ± 1.53	PC-0.11	24 ± 1.53	30 ± 0.58
BNS-1.1	23 ± 1.00	31 ± 1.00	PC-0.14	24 ± 2.08	29 ± 2.52
BNS-1.4	25 ± 1.53	32 ± 0.58	PC-0.17	23 ± 1.00	28 ± 1.00
BNS-1.7	32 ± 1.00	39 ± 0.58	PC-0.20	25 ± 1.15	30 ± 1.15
BNS-2.0	30 ± 0.58	35 ± 1.00			

**Table 4 materials-14-00662-t004:** 5 min and 15 min fluidity of BNS and PC superplasticizers with different dosages.

Superplasticizer/wt%	Fluidity		Superplasticizer/wt%	Fluidity	
	5 min/mm	15 min/mm		5 min/mm	15 min/mm
0	/	/	PC-0.08	110 ± 0.35	/
BNS-0.8	/	/	PC-0.11	115 ± 0.14	/
BNS-1.1	90 ± 0.35	/	PC-0.14	160 ± 0.71	/
BNS-1.4	150 ± 0.00	/	PC-0.17	175 ± 1.41	/
BNS-1.7	205 ± 1.41	100 ± 0.35	PC-0.20	195 ± 0.00	/
BNS-2.0	220 ± 0.71	135 ± 0.14			

Note: / means no fluidity.

**Table 5 materials-14-00662-t005:** Compressive strength of C_4_A_3_S mortars.

Superplasticizer/wt%	Compressive Strength/MPa	
	1 d	7 d
0	30.8 ± 1.45	31.2 ± 1.71
BNS-0.8	32.0 ± 1.77	33.5 ± 1.07
BNS-2.0	36.6 ± 2.21	37.8 ± 1.65
PC-0.08	35.5 ± 1.82	36.5 ± 1.60
PC-0.20	26.3 ± 1.33	28.3 ± 1.13

**Table 6 materials-14-00662-t006:** The amount of ettringite in CSA pastes hydrated with different dosages of superplasticizer.

Superplasticizer/wt%	The Amount of Ettringite/wt%		
	2 h	7 d	28 d
BNS-0.8	26	36	37
BNS-2.0	31	36	37
PC-0.08	24	37	35
PC-0.20	31	37	35

**Table 7 materials-14-00662-t007:** The amount of ettringite in C_4_A_3_S pastes hydrated with different dosages of superplasticizer.

Superplasticizer/wt%	The Amount of Ettringite/wt%		
	2 h	7 d	28 d
0	27	39	39
BNS-0.8	28	40	41
BNS-2.0	30	41	43
PC-0.08	28	40	44
PC-0.20	27	36	39

## Data Availability

The data presented in this study are available in this article.

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
