# Peer review of "Effect of Naphthalene-Based Superplasticizer and Polycarboxylic Acid Superplasticizer on the Properties of Sulfoaluminate Cement"

_materials, 2021, doi:10.3390/ma14030662_

Round 1

Reviewer 1 Report

This article is very interesting but several things must be presented in better way:

  1. Line 13 - put full name of the CSA
  2. Description of an abstract is to general. Authors should put the best results with quantification of the studied quantities
  3. Introduction: (2) formula – 2AH3 was omitted.
  4. 1 Materials. What was the parameters of the powders? How it was mixed?
  5. 1 there is no any comments to the Fig. 1.
  6. 3 Sample preparation. What was the value of the porosity? And Please, put processing steps. There is no information about sample compaction.
  7. 2 Fluidity. Lines from 6 to 11 – the description is unclear.
  8. 2 Fluidity. Line 15: “…new hydration products…” what kind of new product it is?
  9. 3 Compressive strength. Fig. 2. On this two plots we have dots (which are real results) and lines (which are artificially inserted) between them what suggest, that Compressive strength is changed with dosage, in kind of “discontinuous”. Authors should put trend lines or delete “sharp lines”.
  10. 3 Compressive strength. Lines from 27 to 31. – the description is unclear. In this section we have two, very long sentences.
  11. What was the volume of the samples vc time of ettringite grew? Are there any cracks appeared?
  12. This part is to general. There is lack information about value. Author should put results not only general sentences.

Author Response

Dear reviewer,

Thank you very much for your attention and comments on our manuscript entitled "Effect of naphthalene-based superplasticizer and polycarboxylic acid superplasticizer on properties of sulfoaluminate cement".

We have modified the manuscript according to your friendly suggestions. And detailed response to you is attached below. Please see the attachment for revised manuscript.

Sincerely yours,

Guoxin Li

Point 1: Line 13 - put full name of the CSA.

Response 1: Thanks for your remind. CSA is abbreviation of sulfoaluminate cement and it was put in Line 13 of page 1.

Point 2: Description of an abstract is to general. Authors should put the best results with quantification of the studied quantities.

Response 2: Thanks for your suggestion. The abstract was refined as follow.

In order to study what effect and reason of superplasticizers on the setting time, the fluidity, and the compressive strength of calcium sulfoaluminate cement (CSA). Naphthalene-based superplasticizer (BNS) and polycarboxylic acid superplasticizer (PC) were selected to interact with CSA pastes and ye’elimite respectively. X-ray diffraction (XRD), Thermogravimetric (TG) and scanning electron microscopy (SEM) analytical methods were used to investigate the class, amount, and microstructure of the CSA pastes and ye’elimite pastes hydration products under the effect of superplasticizers. The results showed that the addition of BNS can promote ettringite generation and then improved the early compressive strength. As the addition of BNS increasing from 0.8 wt % to 2.0 wt%, the initial setting time prolonged 10 min; the final setting time prolonged 7 min; the 5 min fluidity improved from no fluidity to 220 mm. However, as the addition of PC increasing from 0.08 wt % to 0.20 wt%. the setting time of the PC just changed within 3 min; the 5 min fluidity increased from 110 mm to 195 mm and no 15 min fluidity at all. Observed by SEM, it can be explained that generated ettringite under the addition of PC was layered lacking bonding, and its morphology changed from rod-like to flake-like, leading to a decrease in early compressive strength.

Point 3: Introduction: (2) formula – 2AH3 was omitted.

Point 4: 4 Materials. What was the parameters of the powders? How it was mixed?

Point 5: 1 there is no any comments to the Fig. 1.

Point 6: 3 Sample preparation. What was the value of the porosity? And Please, put processing steps. There is no information about sample compaction.

Point 7: 2 Fluidity. Lines from 6 to 11 – the description is unclear.

5 min and 15 min fluidity of BNS and PC superplasticizers with different dosages are shown in Table 4. The pastes had no 5 min and 15 min fluidity without superplasticizer. When added 0.8 wt% BNS, the pastes had no fluidity at 5 min and 15 min all. With the amount of BNS increased from 1.1 wt% to 2.0 wt%, the 5 min fluidity improved from only 90 mm to 220mm. However, the CSA pastes had no 15 min fluidity untill the BNS dosage increased to 1.7 wt%. The fluidity of 15 min increased from 100 mm to 135 mm, and the 5 min fluidity increased from 205 mm to 220 mm when the BNS dosage grew from 1.7 wt% to 2.0 wt%. The fluidity loss between 5 min and 15 min decreased. It is also can be seen in the Table 4, the 5 min fluidity of PC superplasticizer increased from 110 mm to 195 mm when the dosage added from 0.08 wt% to 0.20 wt%. There was no fluidity of 15 min at all dosages. Therefore, the fluidity loss between 5 min and 15 min was keeping increasing at PC dosage ranges.

Point 8: 2 Fluidity. Line 15: “…new hydration products…” what kind of new product it is?

Point 9: 3 Compressive strength. Fig. 2. On this two plots we have dots (which are real results) and lines (which are artificially inserted) between them what suggest, that compressive strength is changed with dosage, in kind of “discontinuous”. Authors should put trend lines or delete “sharp lines”.

Point 10: 3 Compressive strength. Lines from 27 to 31. – the description is unclear. In this section we have two, very long sentences.

The system of hydration of CSA and C4A3 mortars all would be more homogeneous because more superplasticizer is adsorbed on the surface of cement particles. And then the dispersion of BNS superplasticizer improved and the system generated more ettringite as TG results shown in Table 6 and Table 7. The system with more ettringite showed higher compressive strength. In addition, research showed that ettringite precipitation predominantly occurs via the production of more individual surface-bound needles rather than the existing needles increasing in thickness and length [31]. Hence, the precipitation and growth method of ettringite also need to be considered to study properties changes of hydration system.

Point 11: What was the volume of the samples vc time of ettringite grew? Are there any cracks appeared?

Response 11: Thanks for your question. In this paper, only Thermogravimetric (TG) analysis were used to measured the amount of ettringite changing with the time and superplasticizers dosage. There were no obvious cracks and volume changes by visual inspection in CSA mortar samples at all ages.

Point 12: This part is to general. There is lack information about value. Author should put results not only general sentences.

Response 12: Thanks for your suggestion. The main results had been refined as follow. It also could be found in Line 1-13 of page 16.

(1)As the addition of BNS increasing from 0.8 wt % to 2.0 wt%, the initial setting time prolonged 10 min; the final setting time prolonged 7 min. However, as the addition of PC increasing from 0.08 wt % to 0.20 wt%, the setting time of the PC just changed within 3 min.

(2)The 5 min fluidity improved from no fluidity to 220 mm as the addition of BNS increasing from 0.8 wt % to 2.0 wt%. As the addition of PC increasing from 0.08 wt % to 0.20 wt%, the 5 min fluidity increased from 110 mm to 195 mm and no 15 min fluidity at all.

(3)The addition of BNS can promote ettringite generation and then improved the early compressive strength. Moreover, the 2h amount of ettringite in CSA pastes increased 5 wt% when the PC dosage added from 0.08 wt % to 0.20 wt%, which caused decrease in early compressive strength instead.

(4)The properties of sulfoaluminate cement were affected by the changes in morphology of ettringite, which results from the addition of BNS and PC. The morphology of ettringite observed by SEM changed from rod-like to flake-like under the addition of PC. The generated flake-like ettringite stacked lacking bonding, causing a decrease in compressive strength.

Reviewer 2 Report

In the present paper, the effects of the naphthalene-based superplasticizer and polycarboxylic acid superplasticizer on the setting time, fluidity, and compressive strength of calcium sulfoaluminate cement are studied.

Some points that need revision:

  1. Generally, all abbreviations used in the abstract and conclusion part should be explained i.e. CSA should be described in the abstract firstly.
  2. The conclusion part should be clearer and more focused on the obtained main results.
  3. What is the reproducibility of the experiments studying the effect of different parameters?

Author Response

Dear reviewer,

Thank you very much for your arduous work and instructive advice on our manuscript entitled "Effect of naphthalene-based superplasticizer and polycarboxylic acid superplasticizer on properties of sulfoaluminate cement".

We have modified the manuscript according to your friendly suggestions. And detailed response to you is attached below. Please see the attachment for revised manuscript.

Sincerely yours,

Guoxin Li

Point 1: Generally, all abbreviations used in the abstract and conclusion part should be explained i.e. CSA should be described in the abstract firstly.

Response 1: Thanks for your remind. CSA is abbreviation of sulfoaluminate cement and it was refined in Line 13 of page 1

Point 2: The conclusion part should be clearer and more focused on the obtained main results.

Response 2: Thanks for your suggestion. The conclusion part had been refined and main results were displayed as follow. It also could be found in Line 1-13 of page 16.

(1)As the addition of BNS increasing from 0.8 wt % to 2.0 wt%, the initial setting time prolonged 10 min; the final setting time prolonged 7 min. However, as the addition of PC increasing from 0.08 wt % to 0.20 wt%, the setting time of the PC just changed within 3 min.

(2)The 5 min fluidity improved from no fluidity to 220 mm as the addition of BNS increasing from 0.8 wt % to 2.0 wt%. As the addition of PC increasing from 0.08 wt % to 0.20 wt%, the 5 min fluidity increased from 110 mm to 195 mm and no 15 min fluidity at all.

(3)The addition of BNS can promote ettringite generation and then improved the early compressive strength. Moreover, the 2h amount of ettringite in CSA pastes increased 5 wt% when the PC dosage added from 0.08 wt % to 0.20 wt%, which caused decrease in early compressive strength instead.

(4)The properties of sulfoaluminate cement were affected by the changes in morphology of ettringite, which results from the addition of BNS and PC. The morphology of ettringite observed by SEM changed from rod-like to flake-like under the addition of PC. The generated flake-like ettringite stacked lacking bonding, causing a decrease in compressive strength.

Point 3: What is the reproducibility of the experiments studying the effect of different parameters?

The compressive strength were tested six times and then calculated averages and standard deviations, and the error bars of compressive strength had been displayed in Figure 2 and standard deviations of C4A3 mortars compressive strength had been added in Table 5.

Reviewer 3 Report

Manuscript ID: materials-1084710  

The submitted manuscript concerns the influence of two superplasticizers (BNS and PC) on the selected properties of sulfoaluminate cement. The article is well thought and the contained research is meaningful and important, especially in the aspect of the future potential applications of cement. The overall impression is high and I think the articles will be greatly interesting to the readers.  

However, minor adjustments are required:
[1] Table 3 and 4 are redundant, delete them, and add the content of the superplasticizers in the text.
[2] Explain the choice of superplasticizer content.
[3] In sections 2.4.1, 2.4.2, 2.4.4, and 2.4.5 add information about devices/apparatus used in research (model and manufacturer).
[4] What was the thickness of the sputtering gold layer for SEM analysis?
[5] In the test methods sections add the number of repetitions during the measurements/tests and also add this information in Tables and Figures (n=x). [6] The standard deviation should be added to the results presented in the Tables (+/- SD)
[7] You have stated ''Current researches generally show that this loss of fluidity over time is mainly caused by cement hydration". Add references.
[8] Discuss in more detail the sentence "Crystal structure lacking bonding show decrease in compression strength.

Author Response

Dear reviewer,

Thank you very much for your attention and comments on our manuscript entitled "Effect of naphthalene-based superplasticizer and polycarboxylic acid superplasticizer on properties of sulfoaluminate cement".

We have modified the manuscript according to your friendly suggestions. And detailed response to you is attached below. Please see the attachment for revised manuscript.

Sincerely yours,

Guoxin Li

Point 1: Table 3 and 4 are redundant, delete them, and add the content of the superplasticizers in the text.

Response 1: Thanks for your remind. Table 3 and Table 4 were deleted and the content of the superplasticizers were added in the text Line 5-7 of page 4.

Point 2: Explain the choice of superplasticizer content.

Response 2: To ensure CSA pastes has good workability and no serious bleeding through testing many times the state of CSA pastes mixed with different dosage of superplasticizers. And then the suitable dosage range of BNS was determined to increase from 0.8 wt% by 0.3 wt% to 2.0 wt%; the suitable dosage range of PC was determined to increase from 0.08 wt% by 0.03 wt% to 0.20 wt%. The choice of superplasticizer content was explained and attached in Line 4-5 of page 4.

Point 3: In sections 2.4.1, 2.4.2, 2.4.4, and 2.4.5 add information about devices/apparatus used in research (model and manufacturer).

Response 3: Thank you for your suggestion. The information about devices/apparatus used in research was added in sections 2.4.1, 2.4.2, 2.4.4, and 2.4.5 respectively.

Vicat apparatus is provided by Wuxi Building Material Instrument Machinery Factory, China.

Universal testing machine is provided by Wuxi Xiyi Building Material Instrument Factory, China.

Vacuum dryer with modle-DZF is provided by Beijing Kewei Yongxing Instrument Co., Ltd, Beijing, China.

Rikagu X-ray diffractometer with model-D/MAX-3C is made in Japan.

Thermogravimetric analyzer with modle-SDTQ 600 is provided by Netzsch, Germany.

Ion sputterer with modle-E-1045 is provided by Hitachi, Janpan.

Point 4: What was the thickness of the sputtering gold layer for SEM analysis?

Response 4: All samples for SEM analysis were sputtered by gold layer for 45 s. The gold target sputtering rate of ion sputter (Modle-E-1045, Hitachi, Janpan) is 35 nm/min. Therefore, the thickness of the sputtering gold layer is about 26 nm.

Point 5: In the test methods sections add the number of repetitions during the measurements/tests and also add this information in Tables and Figures (n=x).

Response 5: Thanks for your suggestion. Every setting time sample should be tested three times and then averaged and standard deviations had been marked in Table 3. The compressive strength were tested six times and then calculated averages and standard deviations, and the error bars of compressive strength had been displayed in Figure 2.

Point 6: The standard deviation should be added to the results presented in the Tables (+/-SD)

Response 6: The standard deviation of setting time, fluidity and compressive strength of C4A3 mortars were added in Table 3, Table 4 and Table 5 respectively.

Point 7: You have stated ''Current researches generally show that this loss of fluidity over time is mainly caused by cement hydration". Add references.

Response 7: Thanks for your remind.  We mean that the hydration of cement increased over time and generated more crystal substances such as ettringite overlapping with other binders. The solid substances increased and then the fluidity of system will be decreased. Therefore, the loss of fluidity over time increased. And this part also had been explained in reference 1, 2 and 5. Now we added quotes in this part.

1. Telesca, A.; Marroccoli, M.; Pace, M. L.; Tomasulo, M.; Valenti, G. L.; Monteiro, P. J. M. A hydration study of various calcium sulfoaluminate cements. Concr. Compos. 2014, 53, 224-232.

2. García-Maté, M.; De la Torre, A. G.; León-Reina, L.; Aranda, M. A. G.; Santacruz, I. Hydration studies of calcium sulfoaluminate cements blended with fly ash. Concr. Res. 2013, 54, 12-20.

5. Irico, S.; Gastaldi, D.; Canonico, F.; Magnacca, G. Investigation of the microstructural evolution of calcium sulfoaluminate cements by thermoporometry. Concr. Res. 2013, 53, 239-247.

Point 8: Discuss in more detail the sentence "Crystal structure lacking bonding show decrease in compression strength.

Response 8: The addition of PC superplasticizer changed the morphology of ettringite, which is verified in Fig. 8 (d) and (e). A large amount of flake-like ettringite layered in Fig. 8 (e). Firstly, in the previous XRD analysis of Fig. 4 (c) and (d), it was found that the addition of PC did not produce any new crystalline material except ettringite. Secondly, in the TG analysis, it was found that the amount of ettringite was not reduced for the dosage of PC. According to Yan’s SEM research on ettringite [33], different crystal morphology of ettringite had different properties Therefore, the decrease in strength is attributed to the change in the shape of ettringite. The formation and growth of ettringite has been changed from rod-like to flake-like under the addition of PC. Ettringite with stacking morphology led to decreases in compressive strength.

  1. Yan, P.Y., Yang, W.Y. SEM research on expansion mechanism of ettringite. Journal of Chinese Electron Microscopy Society. 1994, 4, 297-300.

Round 2

Reviewer 3 Report

The manuscript (materials-1084710) has been revised in line with all my comments. The authors have improved it accurately and correctly. Hence, I would recommend accepting the article in present form.